# Detection of Key Organs in Tomato Based on Deep Migration Learning in a Complex Background

**Jun Sun \*, Xiaofei He, Xiao Ge, Xiaohong Wu, Jifeng Shen and Yingying Song**

School of Electrical and Information Engineering of Jiangsu University, Zhenjiang 212013, China;
hxf19930620@sina.com (X.H.); 18352867556@163.com (X.G.); wxh419@ujs.edu.cn (X.W.);
shenjifeng@ujs.edu.cn (J.S.); m18839381604@163.com (Y.S.)
\* Correspondence: sun2000jun@ujs.edu.cn; Tel.: +137-7554-4650

**Abstract:** In the current natural environment, due to the complexity of the background and the high similarity of the color between immature green tomatoes and the plant, the occlusion of the key organs (flower and fruit) by the leaves and stems will lead to low recognition rates and poor generalizations of the detection model. Therefore, an improved tomato organ detection method based on convolutional neural network (CNN) has been proposed in this paper. Based on the original Faster R-CNN algorithm, Resnet-50 with residual blocks was used to replace the traditional vgg16 feature extraction network, and a K-means clustering method was used to adjust more appropriate anchor sizes than manual setting, to improve detection accuracy. The test results showed that the mean average precision (mAP) was significantly improved compared with the traditional Faster R-CNN model. The training model can be transplanted to the embedded system, which lays a theoretical foundation for the development of a precise targeting pesticide application system and an automatic picking device.

**Keywords:** object detection; tomato organ; K-means clustering; Soft-NMS; migration learning; convolutional neural network; deep learning

## 1. Introduction

Tomato is native to South America, and it has a cultivation history of more than 100 years in China. It is one of the most popular vegetables, with health effects of lowering blood pressure, slowing down aging, increased slimming, and vitamin supplementation [1].

Due to the wide planting area, high yield, and short maturity, tomato is difficult to store. The picking of tomato is a labor-intensive and time-consuming operation. Therefore, the development of an automatic picking robot for tomato is particularly important. The technology of object detection based on image processing is the most basic and important link in the research of the picking robot. It is a typical object recognition task that detecting the key organs (flower and fruit) of tomato from images of plants. Data from the Internet shows that immature green tomato contains alkaloids, which may cause poisoning after eating, while mature red tomato does not contain alkaloids. Therefore, it is significantly important to detect the key organs (tomato flower, immature green tomato, and mature red tomato) accurately and immediately. This will provide a theoretical basis for the detection of pests and diseases, targeted pesticide application systems, and the development of the picking robot [2].

Traditional tomato recognition methods mostly extract information on color, shape, and some shallow features (the visual features of the bottom layer, such as texture features and the colors and shapes mentioned earlier), then classifiers are used to detect and recognize the tomato. Zhao et al. [3] used the combination of Harr features (a Haar feature is a kind of feature that reflects the change of gray level of the image, and the pixel difference is calculated by the module [4]) and an Adaboost classifier,

to identify tomatoes. The recognition rate of mature tomato in the test set was 93.3%, but it was easy to misidentify tomatoes in the scene of illumination conversion; Jiang et al. [5] utilized the difference of color characteristics between mature tomatoes and background to identify red mature tomatoes by threshold segmentation. However, the recognition rate of immature green tomatoes in which the color is similar to the leaf, is not high; Zhang et al. [6] used gray transformation to extract the edge features of the background region, then segmented the tomato from the background by a fitting curve, and obtained the three-dimensional coordinates of the target by using the principle of stereo vision imaging; Yamamoto et al. [7] extracted the color characteristics of leaves, stems, and backgrounds, then constructed a decision tree through a regression tree classifier, and extracted the pixels of tomato fruits by blob pixel segmentation, to identify tomatoes. However, this method had a poor identification rate under the complex backgrounds.

At present, most traditional object recognition and detection methods are based on shallow feature extraction to detect and identify tomato organs. The overlapping and occlusion problems between tomatoes under a complex background cannot be easily solved, and the time cost for feature extraction is relatively expensive, and the applicability is not strong.

In recent years, with the emergence of deep learning, convolutional neural networks (CNN) can extract hierarchical features by using unsupervised or semi-supervised feature learning, which has stronger generalization than artificial features. It can not only learn shallow semantic information, but also learn deep abstract information [8]. In the field of generalized recognition (such as object classification [9], object detection [10] and segmentation [11]), Faster R-CNN (faster regions with convolutional neural network features) [12] and YOLO (you only look once) [13] models have been widely used, and have achieved good results. Bargoti et al. [14] constructed an image segmentation processing framework by using orchard image data. Multi-scale multilayer perceptron (MLP) and convolutional neural network were include in the framework. The image data captured from the network was extended by context information, as well as some of the appearance changes and category distributions that are observed in the data. Finally, Watershed Segmentation (WS) and Circular Hough Transform (CHT) processing algorithms were used for fruit detection and counting. However, this method was more complicated; Sa et al. [15] constructed the Faster R-CNN model by pixel level superposition of RGB and near-infrared images, then the model was used for fruit detection by migration learning, but the number of model parameters was too large and was not suitable for real-time detection; Zhou [16] proposed a Fast R-CNN model that is based on double convolutional chains. The feature extraction network based on vgg16 was trained by fusing RGB and grayscale feature images of the tomato, and then Fast R-CNN was initialized with the parameters to identify flowers, fruits, and stems of the tomato. However, this method did not solve the problem of occlusion of tomatoes by stems and leaves. The above literature proves that CNN can not only extract the shallow texture and color of key organs of tomato automatically, but it can also learn deeper abstract features. This will improve the detection accuracy of tomato flowers and fruits, and reduce the cost of feature extraction, which is more robust to detection and identification in a complex environment. Thus, it is feasible to use deep learning methods to detect and identify tomato organs. However, the accuracy of these models is not high under the conditions of overlap and occlusion, because there will be overlap and occlusion between the tomatoes, and the stems and leaves of the tomato will also cover the fruit. Additionally, the anchor size used in the above method was set artificially by the target in the VOC 2007 (Pascal Visual Object Classes Challenge 2007) dataset, without updating according to the actual size of the key organs of tomato, which will have a certain impact on the recognition accuracy.

Due to the fixed anchor size, overlap, and occlusion of the stems and leaves acting on the fruit, the recognition rate of tomato is low. Thus, an improved Faster R-CNN model was proposed in this paper. Firstly, the K-means clustering was used to obtain the anchor size that was suitable for the key organs of tomato. Secondly, residual blocks were used to replace the basic feature extraction network of the original model in order to improve the detection and recognition accuracy of tomato key organs. Finally, Soft-NMS (soft-non maximum suppression) was used to attenuate the bounding

boxes of the tomato organ instead of completely removing them, which can solve the above problems to a certain extent.

## 2. Materials and Methods

### 2.1. Data Sources

A total of 5624 RGB images of tomato flowers, mature red tomatoes, and immature green tomatoes were collected from the agricultural digital greenhouse of Jiangsu University with a high-definition camera (Dataset A). These pictures were taken at 9:00 a.m., 12:00 a.m., and 3:00 p.m., July–August, 2018. The camera is a CMOS (Complementary Metal Oxide Semiconductor) sensor, which comes with the Honor 9 mobile phone. All images were 1080 × 1920 pixels and have a resolution of 96 dpi. In order to prevent the poor performance of the model caused by insufficient diversity of training samples, the following measures were taken during the process of image acquisition: considering the difference in imaging results caused by different greenhouse ambient light conditions, images were collected in sunny weather and cloudy days, respectively; in the process of sampling, different forms and occlusions of tomato organs were taken into account, and fruits with different maturity were photographed from multiple angles to increase the diversity of samples. In this paper, labelImg software (this can be obtained from this website [17]) was used to label the tomato flowers, green tomatoes, and red tomatoes in the dataset. After each image was annotated, a corresponding xml file containing the category and location information of the target, similar to the dataset format of the PASCAL visual object classes challenge 2007 [18], was generated. A large amount of Internet information showed that a lot of data was required to train the models by using deep learning methods. Combined with the instruction manual of the deep learning framework keras, a Python script was used to augment a small number of sample maps, including random flip (horizontal, vertical), transform angle (0~180), random scaling of the original image scaling factor (1~1.5) etc. [19]. The total number of expanded samples was 8929 (Dataset B, as is shown in Figure 1), and then the expanded set was randomly divided into 4:1 ratios between the training set and the test set. The expanded training set was used to train the model. All parameter settings and discussions in this paper only pertain to Dataset B.

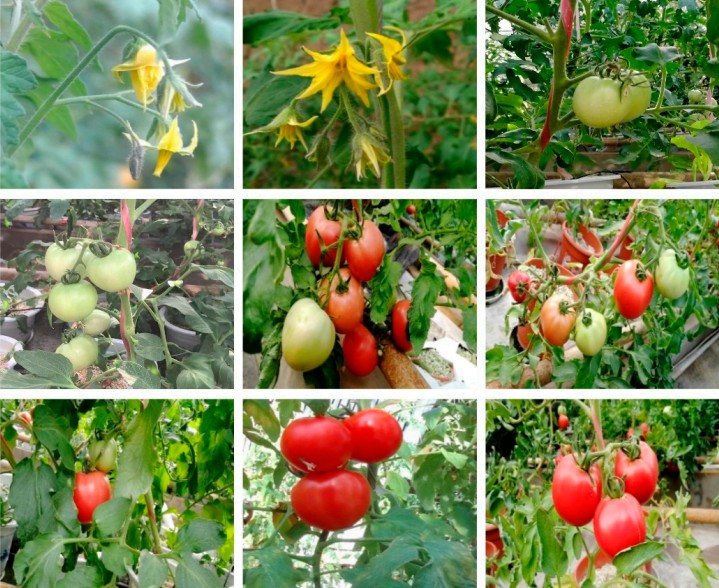

**Figure 1.** Examples of key organs of tomato.

### 2.2. Structure of the Detection Model

CNN includes the convolutional layer, pooling layer, and fully connected layer. The convolutional layer uses semi-supervised feature learning and a hierarchical feature extraction efficient algorithm

to extract image abstract features. It can automatically extract and reduce the dimension of input images, and it has stronger generalization than the human-set features. The fully connected layer mainly performs image classification based on the extracted features. The neurons in the convolutional layer extract the primary visual features of the image by using a local receptive field, and they reduce the network parameters by sharing the weights. The pooling layer not only reduces the dimension of the features, but also realizes the invariance of displacement, scaling and distortion. The convolutional layer and pooling layer in CNN usually appear alternately, and the activation unit is set to realize the nonlinear transformation, which accelerates the convergence rate of the network.

The traditional Faster R-CNN object detection algorithm can be divided into two parts. One is the feature extraction layer and the other is the region proposal network (RPN). Vgg16 was selected as the basic feature extraction network for image feature extraction and classification. The network consists of eight convolutional layers, five maximum pooling layers, and three fully connected layers, and Re LU (Rectified Linear Unit) was used as the activation function. In RPN, an arbitrary scale image is taken as input and outputs a series of object proposals, and each proposal has an object score. In order to generate region proposals, a slide is made on the feature map of the last convolutional layer of vgg16, then some ROIs (region of interests) are generated and multiple region proposals are predicted at the same time in each sliding window position [20].

### 2.3. An Improved Feature Extraction Network

Deep convolutional neural network has made great progress in image recognition compared with traditional methods. As the number of network layers deepens, the convolutional layer can learn deeper abstract features, and the related research also shows that the extracted features can be enriched by increasing the number of network layers. Simon et al. [21] demonstrated that the accuracy of recognition rises with the increase of the depth of the network. However, the simple stacked convolutional layer cannot train the network smoothly, due to the gradient explosion when propagating backward. Although the relevant literature has shown that the deep network can be trained through batch normalization [22] and dropout [23], the problem of precision decline after a certain iteration remains. In order to break through the problems of the precision degradation of the deep network, and the limitation of network depth, He et al. proposed a deep residual resnet model by using the concept of identity mapping. This method solved the degradation problem by fitting a residual map with a multi-layer network [24]. For a stacked layer structure (stacked by several layers), if the residual is zero, then the stacked layer only makes an identity map, but the network performance will not decrease. However, in fact, the residual is not zero, which can enable the stacked layers to learn new features based on the input characteristics, thus having better performance. The basic idea is as follows (as shown in Figure 2):

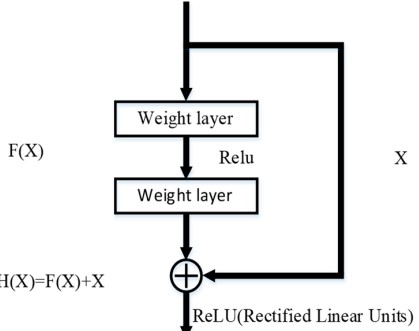

**Figure 2.** Residual learning module. Since direct mapping is difficult to learn, the basic mapping relationship from X to H(x) is no longer being learned, but the difference between the two is learned, which is the residual, and in order to calculate H(x), this residual is simply added to the input.

At this point, the original optimal solution mapping H(x) can be equivalent to F(X) + X, that is, the fast connection implementation in the feed-forward network that is shown in Figure 2. The method of quick connection can be expressed by Formula (1):

$$Y = F(X, \{W_i\}) + W_s X \tag{1}$$

In Formula (1), X represents the input vector of the module, Y represents the output vector of the module, $W_i$ represents the weighting layer parameters, and a linear projection $W_s$ is needed to match the dimensions to ensure the consistency of the input and output dimensions.

ResNet-50 consists of 50 modules with the same structure, as shown in Figure 2. In this paper, the vgg16 feature extraction model was replaced by the 50-layer residual network ResNet-50, to improve the classical Faster R-CNN depth learning model, and the improved detection framework was displayed in Figure 3. Since the time cost of pre-training a resnet model from imagenet is expensive, and the feature extraction performance of ResNet-50 can achieve a good result, the resnet model using 101 layers and 152 layers will not be discussed in this paper.

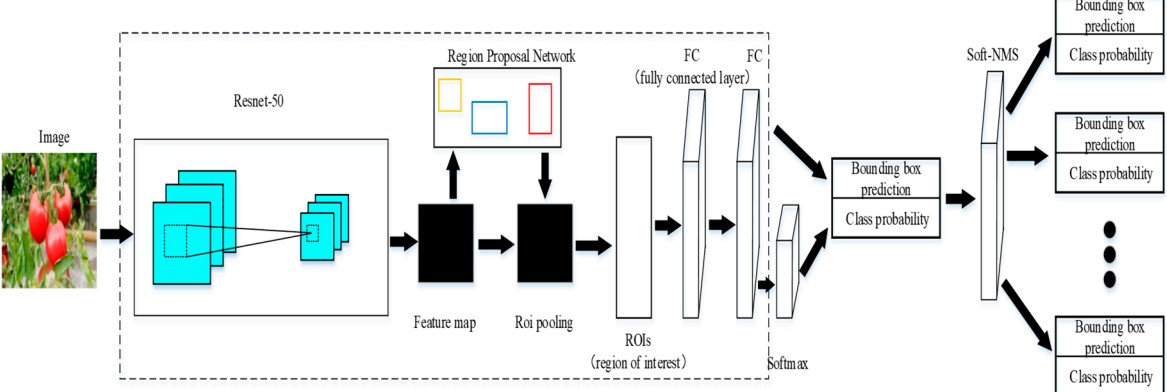

**Figure 3.** Improved framework for the Faster R-CNN model. Firstly, the image is input, and the characteristics of the key organs of tomato are extracted through the feature extraction network Resnet-50, to generate a feature map. Then, ROI pooling is performed after the region proposals are generated on the feature map. At last, the recognition and localization tasks are completed after two fully connected layers.

### 2.4. Use K-Means to Cluster the Appropriate Anchor Size

The purpose of the clustering algorithm (K-Means) is to divide *n* objects into *k* different clusters, according to their respective attributes, so that the similarity of each object in the cluster is as high as possible, and the similarity between clusters is as small as possible. The criterion function that is used for evaluating similarity is the sum of squared errors. The network can learn to properly adjust the box, but if better priors are chosen, it can make the network easier to learn predictive detection. Instead of manually selecting priors, the k-means clustering algorithm is used to traverse the training set bounding box and automatically find good priors.

The original Faster R-CNN used nine kinds of anchor size; thus, the value of *k* must be set to 9 to cluster, in order to obtain the same number for the new anchor size. Since the convolutional neural network has translation invariance and the positions of the anchor boxes are fixed by each grid, it is only necessary to calculate the widths and heights of the anchor boxes by the k-means. If a standard k-means with Euclidean distance is used, a larger box will produce more errors than the smaller box. However, the most important thing is to be able to obtain priors that lead to good IOU (Intersection

over Union) scores, which are independent of the size of the box. Therefore, the new formula below is used to measure the distance:

$$d(box, centroid) = 1 - \text{IOU}(box, centroid) \tag{2}$$

In the above formula, IOU represents the coincidence of the prediction box and the label box. When the anchor boxes are calculated, the abscissa and ordinates of the center points of all boxes will be set to zero. Thus, all the boxes are in the same position, which is convenient for calculating the similarity between boxes by the new distance formula. As a k-means algorithm was used to traverse the label information of all marked tomato key organs, the more appropriate anchor sizes suitable for the dataset were obtained according to the sizes of the manually labeled bounding boxes. When the clustering algorithm was run, new anchor sizes were generated (sizes from 69 pixels to 247 pixels), and the original anchor sizes, $8 \times 16$, $16 \times 16$, $32 \times 16$ pixels were updated to $4 \times 16$, $8 \times 16$, $16 \times 16$ pixels, respectively. Thus, RPN could better detect some smaller sized targets when generating region proposals. This can avoid missing detection and improve recognition accuracy. For a more detailed description, please refer to [25].

*2.5. Region Proposal Network*

The core idea of RPN is to directly generate region proposals by using CNN, which is essentially a sliding window [26]. At the center of the sliding window, a total of nine kinds of anchors are generated, corresponding to three basic scales (8, 16, 32) and three aspect ratios (0.5, 1, 2) of the input image. Then the predicted region proposals are sent to two fully connected layers: cls layer and reg layer, which are used for classification and box regression, respectively. Finally, the top 300 region proposals are selected as inputs of Fast R-CNN after sorting according to the score of region proposals.

*2.6. Soft-NMS*

Non-maximum suppression (NMS) is used to suppress elements that are not the maxima, searching for local maxima. This local area represents a neighborhood with two variable parameters: one is the dimension of the neighborhood, and the other is the size of the neighborhood. Non-maximum suppression is an important part of the object detection process, which generates the detection box based on the object detection score. The detection box with the highest score is selected, while other detection boxes with obvious overlap with the selected detection box are suppressed. This process is continuously recursively applied to the remaining detection boxes [27]. For example, in pedestrian detection, each window will obtain a score after feature extraction and classifier recognition, but the sliding window will also cause many windows to overlap with other windows, so that NMS is needed to select the detection box with the highest score in the neighborhood, and suppress those boxes with low scores.

Soft-NMS has the same algorithmic complexity as traditional NMS. It only needs to make simple changes to the traditional NMS algorithm without adding additional parameters and training, and it can be easily integrated into any object detection process with high efficiency and easy implementation.

The score reset function of the traditional NMS (rescoring function) is calculated as follows:

$$s_i = \begin{cases} s_i, & iou(M, b_i) < N_t \\ 0, & iou(M, b_i) \geq N_t \end{cases} \tag{3}$$

In the above Formula (3), a threshold is used in NMS to determine whether adjacent detection frames are reserved. Among them, $M$ is the bounding box with the highest current score, $b_i$ is the adjacent detection box, $S_i$ is the score of detection box, and $N_t$ is the threshold. When the *iou* of the $M$ and $b_i$ is less than the threshold $N_t$, $S_i$ will not change, otherwise, $S_i$ will be set to zero.

The score reset function of the Soft-NMS (rescoring function) is calculated as follows:

$$s_i = \begin{cases} s_i, & iou(M, b_i) < N_t \\ s_i(1 - iou(M, b_i)), & iou(M, b_i) \geq N_t \end{cases} \tag{4}$$

The score of the adjacent detection box that overlaps with the detection box $M$ by attenuation is an effective improvement to the NMS algorithm. When the *iou* of the $M$ and $b_i$ is less than the threshold $N_t$, $S_i$ will not change; otherwise, $S_i$ equals $S_i(1 - iou(M, b_i))$. The higher the overlap of $M$ is, the more serious the fractional attenuation may be. When the degree of overlap between the adjacent detection box and $M$ exceeds the threshold $N_t$, the detection score of the detection box is linearly attenuated. In this case, the algorithm attenuates the detection score of the non-maximum detection box instead of completely removing it, and it does not attenuate the original detection score of the detection box without overlapping. Therefore, the non-maximum detection box of the tomato under overlapping and occlusion is retained, and the detection precision is improved.

## 3. Model Training

### 3.1. Test Platform

The test operating platform is the Ubuntu 16.04 system, which uses the Tensorflow as a deep learning framework. The computer memory is 32 GB, equipped with Intel® Core™ i7-7700K CPU@4.00GHz × 8 processors, the GPU used is NVIDIA GTX1080Ti, adopting a 16 nm production process, the memory type is GDDR5, and the capacity is 11 GB.

### 3.2. Test Parameter Settings

The mini-batch stochastic gradient descent (SGD) method with momentum factor was used to train the network [28]. The number of the mini-batch was 256, and the momentum factor was set to 0.9. Since the initialization of weights affects the convergence speed of network, the Gaussian distribution (the mean value was zero and the standard deviation was 0.01) was used to initialize the weights of all layers of the network randomly in this paper. All bias of the convolutional layers and fully connected layers were initialized to zero. The same learning rate was adopted for all layers in the network, and the initial learning rate was set to 0.001. During the training process, the current learning rate was reduced by 1/10 step by step, and the regularization coefficient was set to 0.0005. All models were iterated 250,000 times.

### 3.3. Indicators for Model Performance Evaluation

In this paper, the precision–recall (P-R) curve, average precision (AP), and mean average precision (mAP) were used as the evaluation metric for tomato detection [29]. Precision refers to the accuracy. In the field of information retrieval, precision and recall appear together. For a query, a series of goals returned, and the correct rate, refers to the proportion of the relevant targets in the returned results. Suppose true positive (TP): predicts the number of positive classes as a positive class; true negative (TN): predicts the number of negative classes as negative classes; false positive (FP): predicts negative classes as positive ones (false rate); false negative (FN): predicts the positive class as the number of negative classes (missing rate). Then, precision is defined as:

$$Presision = \frac{TP}{(TP + FP)} \tag{5}$$

The recall rate is the ratio of relevant targets in the returned results to all relevant targets, and it is defined as:

$$Recall = \frac{TP}{(TP + FN)} \tag{6}$$

Compared to the graph, in some cases, the specific values can more clearly show the performance of the test model. The average precision (AP) is usually used as a metric, and the calculation formula is:

$$\text{AP} = \int_0^1 p(r)d(r) \tag{7}$$

Among Formula (7), *p* represents *Precision*, *r* represents *Recall*, and *p* is a function of *r*. So, the average precision is equal to the area under the P-R curve, and mAP equals the average of all categories of average precision.

## 4. Experimental Results

In this section, the performance of the proposed method was evaluated and analyzed in three experimental settings: (1) detection performance under different basic anchor size; (2) detection performance under different pooling types of the basic feature extraction network; (3) detection performance under different retention algorithms of the bounding box.

### 4.1. Detection Performance under Different Basic Anchor Sizes

Table 1 shows the model parameter settings and mAP. Each model has run a total of 250,000 times, and has converged. Since anchors were used to predict the bounding boxes, three basic sizes, 8, 16, 32, and three conversion ratios, 0.5, 1, 2 (a total of nine kinds of anchors) were applied to the original Faster R-CNN. The anchor size is equal to the basic size multiplied by 16 pixels. Because the target size of the original voc dataset was very different, if the tomato dataset was used to train the model, some anchors were unreasonable. Therefore, the appropriate anchor size by K-means clustering method was calculated in this paper. The original size was updated to $4 \times 16$, $8 \times 16$, $16 \times 16$ pixels, and the transformation ratio remained unchanged, in order to improve the detection rate of the bounding box. Generally, the threshold of mAP is 0.5, which means that if the coincidence degree is greater than 0.5, the key organs of the tomato are correctly detected. From Table 1, it can be seen that on dataset B, the mAPs of models 2, 4, 6, and 7 are 0.7, 1.6, 0.7, and 1.6 percentage points higher than models 1, 3, 5, and 8, respectively. The above situation indicated that the updated anchor size is more suitable for the dataset of this paper, and that the recognition rate of key organs of tomato is improved. The reason for this phenomenon is that the original anchor size is manually pre-set, and is suitable for the larger target of the voc2007 dataset. However, the dataset in this paper contains targets such as small tomatoes. If the original size is used, small targets will not be detected; thus, K-means need to be used to cluster the appropriate anchors. In order to more intuitively represent mAPs of each model, Table 1 was converted into the form of a chart as Figure 4.

**Table 1.** Model parameter settings and mean average precision (mAP).

| Model NO. | Basic Feature Extraction Network | Model Parameters | | Detection Box Retention Algorithm | Testing Dataset A mAP (%) | Testing Dataset B mAP (%) |
| | | Basic Anchor Size (pixels) | Pooling Type of Basic Feature Extraction Network | | | |
|---|---|---|---|---|---|---|
| 0 | VGG16 | $8 \times 16, 16 \times 16, 32 \times 16$ | max-pooling | NMS | 83.1 | 85.2 |
| 1 | ResNet50 | $8 \times 16, 16 \times 16, 32 \times 16$ | average-pooling | NMS | 85.5 | 86.6 |
| 2 | ResNet50 | $4 \times 16, 8 \times 16, 16 \times 16$ | average-pooling | NMS | 86.0 | 87.3 |
| 3 | ResNet50 | $8 \times 16, 16 \times 16, 32 \times 16$ | max-pooling | NMS | 85.9 | 87.1 |
| 4 | ResNet50 | $4 \times 16, 8 \times 16, 16 \times 16$ | max-pooling | NMS | 86.9 | 88.7 |
| 5 | ResNet50 | $8 \times 16, 16 \times 16, 32 \times 16$ | average-pooling | Soft-NMS | 87.7 | 88.9 |
| 6 | ResNet50 | $4 \times 16, 8 \times 16, 16 \times 16$ | average-pooling | Soft-NMS | 88.3 | 89.6 |
| 7 | ResNet50 | $4 \times 16, 8 \times 16, 16 \times 16$ | max-pooling | Soft-NMS | 89.3 | 90.7 |
| 8 | ResNet50 | $8 \times 16, 16 \times 16, 32 \times 16$ | max-pooling | Soft-NMS | 88.2 | 89.1 |

**Note**: Testing Dataset A represents the original dataset without augmentation, and Testing Dataset B represents the expanded dataset using augmentation. All parameter settings and discussions in this paper are only performed on Dataset B.

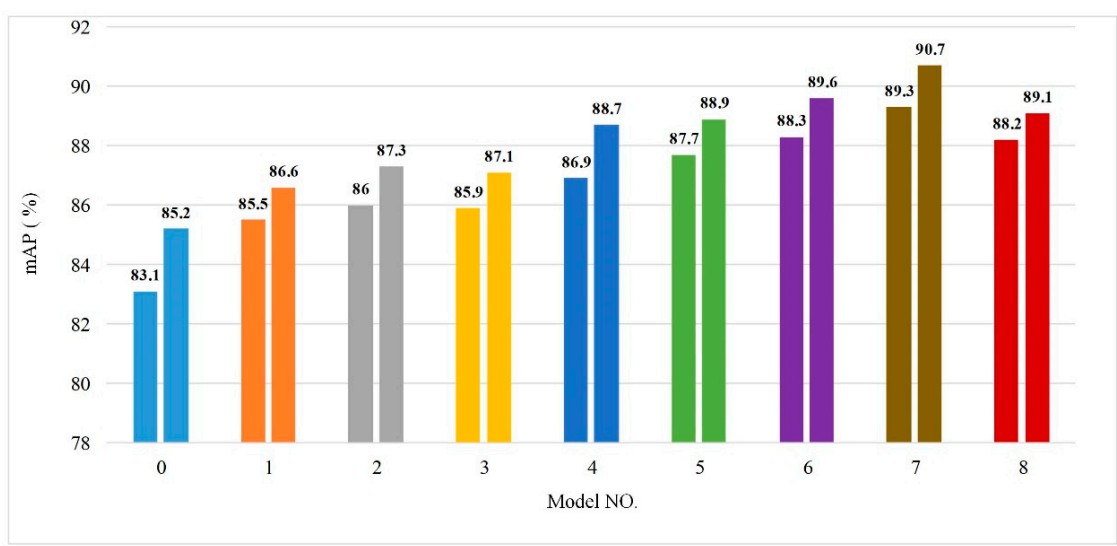

**Figure 4.** The bar chart of mAP for different model parameter settings. Table 1 is visualized to more intuitively show the mean average precision of the models with different parameter settings. As can be seen from Figure 4, the numbers 0–8 represent models 0–8 in Table 1, respectively. The first histogram for each model is Dataset A, and the latter one is Dataset B.

The testing result of the detection model using the original anchor sizes and the updated anchor sizes is shown in Figure 5.

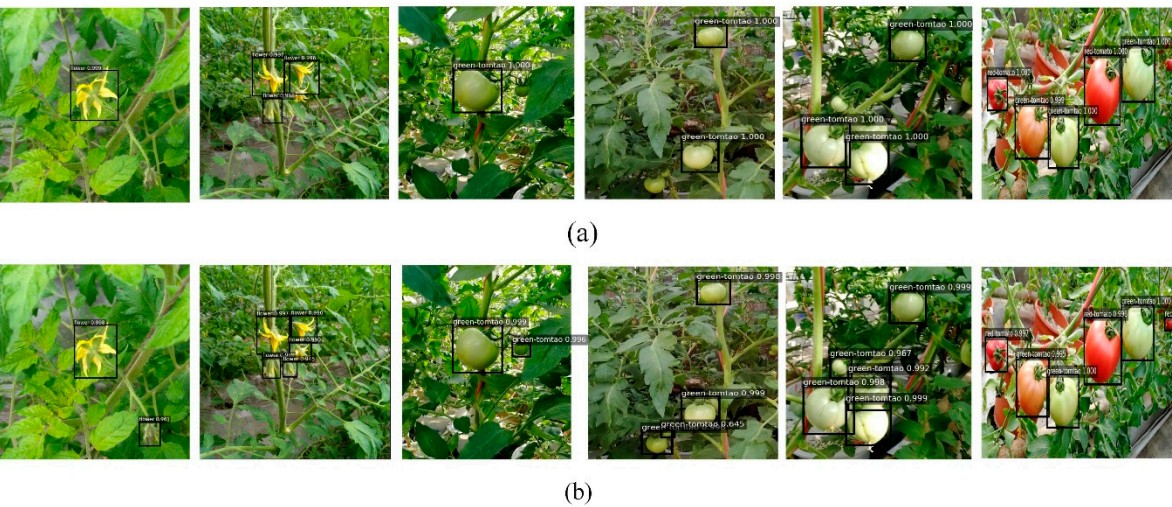

**Figure 5.** Visualization results using different anchor sizes. (**a**) Testing results of model 1 using the original anchor sizes (8 × 16, 16 × 16, 32 × 16 pixels); (**b**) Testing results of model 2 using updated anchor sizes (4 × 16, 8 × 16, 16 × 16 pixels). The reason for this phenomenon is that the original anchor size is manually pre-set, and is suitable for the larger target of the voc2007 dataset. However, the dataset in this paper contains targets such as small tomatoes and flowers. If the original size is used, small targets will not be detected; thus, K-means need to be used to cluster the appropriate anchors.

### 4.2. Detection Performance under Different Pooling Types of the Basic Feature Extraction Network

There are usually two types of pooling: maximum pooling and average pooling. In Table 1, model 3, 4, 7, 8 and model 1, 2, 5, and 6 respectively adopts the maximum and average pooling types. The results show that the effect of maximum pooling is better than the average pooling when other conditions remain unchanged. It can be seen from the figure that the model using the maximum pooling has higher confidence than the model using the average pooled model, and some targets

that are similar to the colors of the background are well recognized. Combining the literature [30] with the Internet data, it can be explained that the effect of average pooling is to average all of the values of the entire feature map. It can reduce the error caused by the increase of estimation variance due to the limited size of the neighborhood, and retains more background information of the image. However, maximum pooling can reduce the deviation of the estimation mean caused by the error of convolutional layer parameters, and retain more texture information. In the object detection task, more texture information may be needed, and the interference information, such as the background, should be reduced as much as possible. So, using maximum pooling in the middle layer of convolution can better preserve texture features, and discard redundant features, which is beneficial to extracting the deep key features of the targets in the image, and improve the recognition accuracy. Figure 6 is a testing result chart using different pooling types.

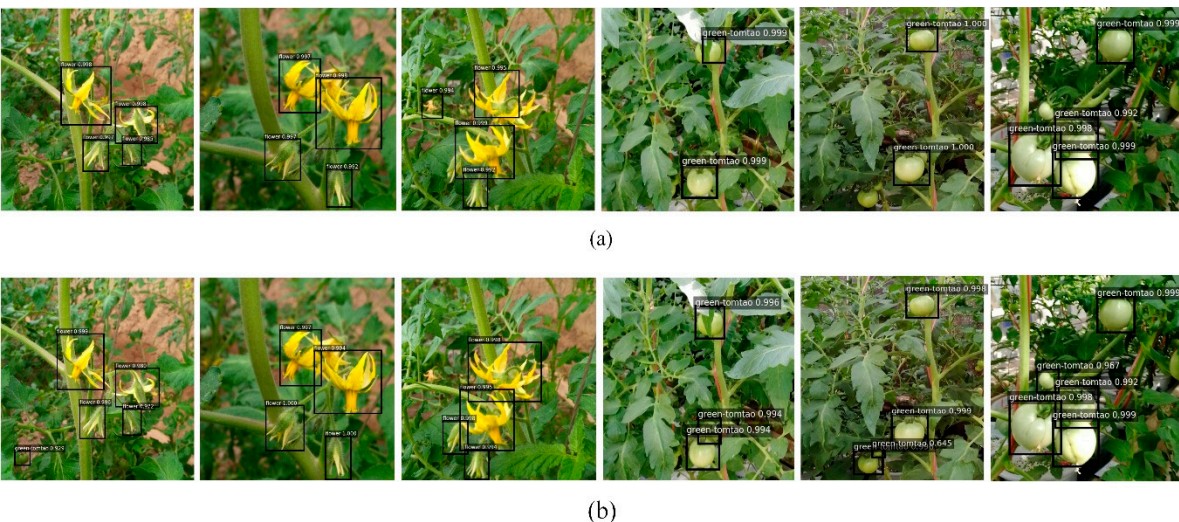

**Figure 6.** Visualization results using different pooling types. (**a**) Testing results of model 2 using average-pooling; (**b**) Testing results of model 4 using max-pooling.

### 4.3. Detection Performance under Different Retention Algorithms of the Bounding Box

It can be seen from Table 1 that the mAP of model 5 is 2.3 percentage points higher than model 1, the mAP of model 6 is 2.3 percentage points higher than model 2, the mAP of model 8 is 2 percentage points higher than model 3, and the mAP of model 7 is 2 percentage points higher than model 4. In order to better demonstrate the effect of the Soft-NMS algorithm on the detection of tomato organs in the case of overlap and occlusion, the detection result was visualized. As is shown from Figure 7, the dataset in this paper has a lot of overlap between tomatoes, and occlusion of stems and leaves. Thus, the optimal model 7 using Soft-NMS is based on the area of overlapping portion to set an attenuation function for the adjacent detection box, instead of completely zeroing its score. This operation can effectively solve the problem of low detection and recognition rates of key organs of tomato under overlapping and occlusion.

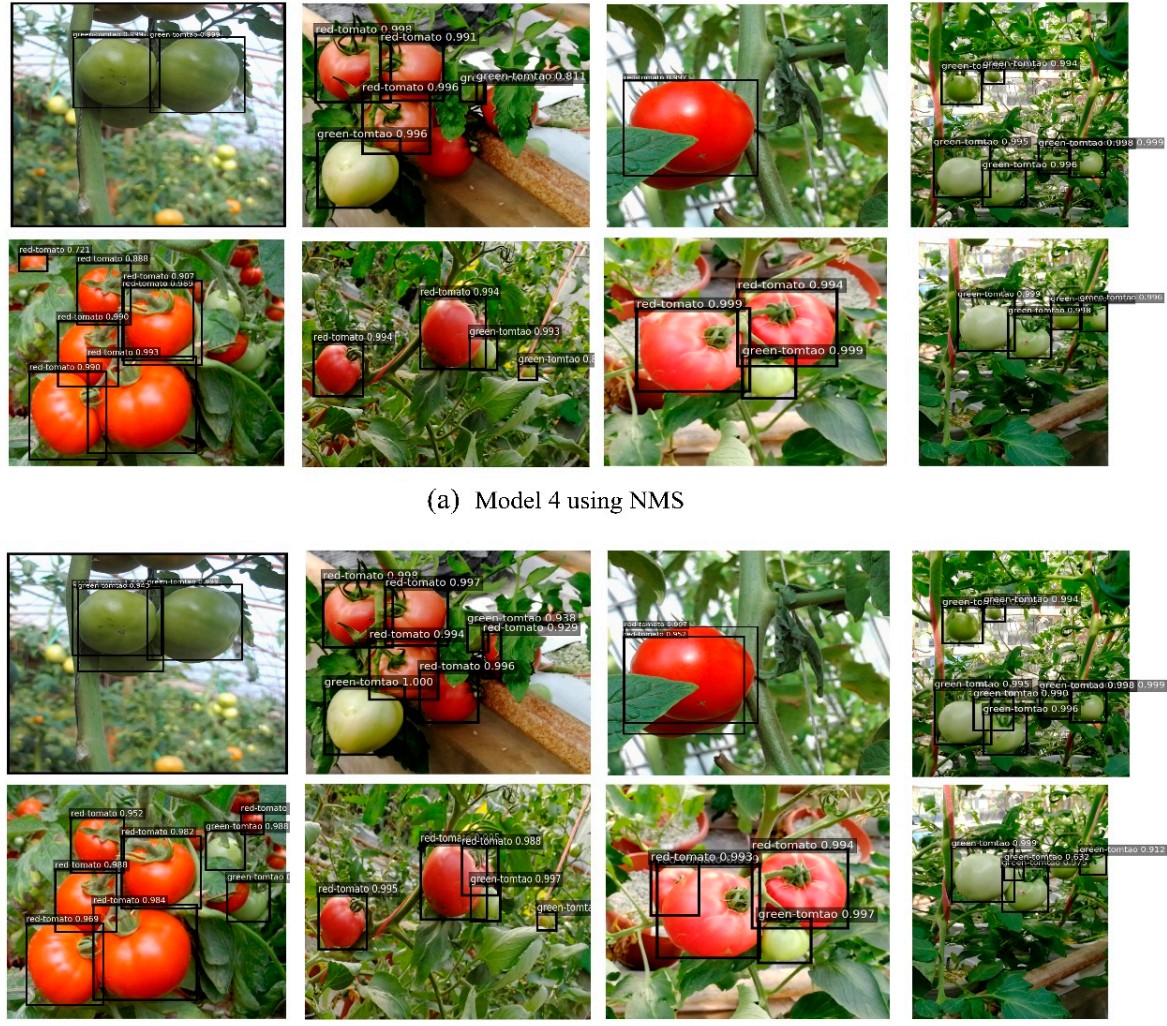

(a) Model 4 using NMS

(b) Model 7 using Soft-NMS

**Figure 7.** Visualization of detection performance in the case of overlap and occlusion. (**a**) Detection of key tomato organs by using a retention algorithm with non-maximum suppression; (**b**) Detection of key tomato organs by using a retention algorithm with Soft-NMS.

## 5. Model Comparison Results

In this paper, three improvements were applied to improve the accuracy of tomato detection. Firstly, resnet-50 with residual blocks was used to replace the traditional vgg16 feature extraction network, then K-means clustering method was used to adjust a more appropriate anchor size than manual setting; finally, Soft-NMS was used to replace the traditional NMS algorithm. The following is a comparison of some results between the improved model 7 and the original model.

### 5.1. Precision and Recall Curve

The performance of the object detection model is usually evaluated by using the recall and precision score curves with the threshold [31]. Figure 8 shows the P-R curve of the original Faster R-CNN and the improved model 7. The area under the curve represents the AP of the class, and the larger the area, the higher the AP. It can be seen from the P-R curve that the detection accuracy of model 7 is higher, and the performance is better. For a detailed introduction to the P-R curve, we can refer to [30].

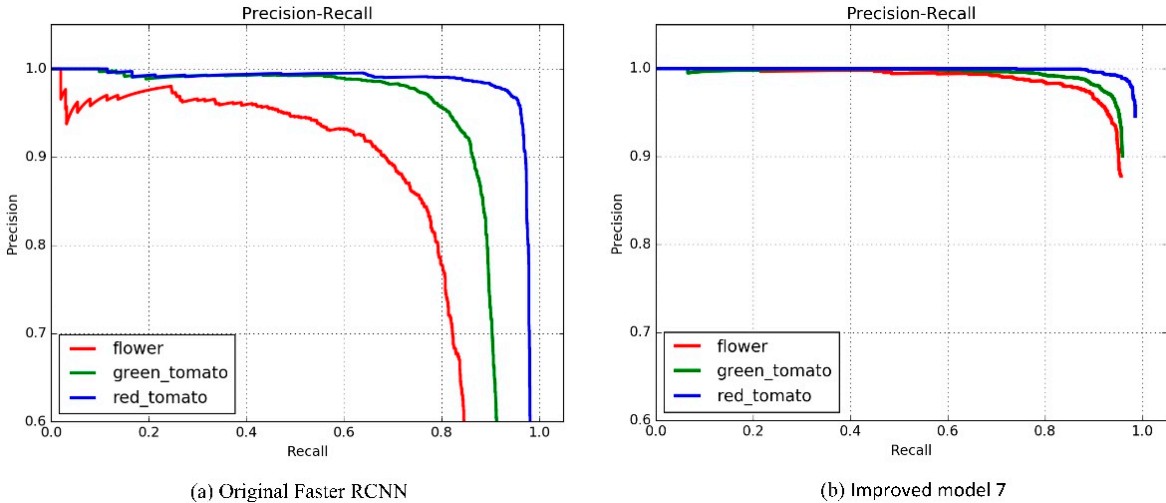

**Figure 8.** Precision and recall curve of the model. (**a**) Precision-recall curve of the original Faster R-CNN model; (**b**) Precision-recall curve of the improved model 7.

*5.2. Average Precision and Mean Average Precision*

It can be seen from Table 2 that the AP of the flowers, immature green tomatoes, and mature red tomatoes in improved model 7 were 13.7, 2.4, and 0.5 percentage points higher than that of the original Faster R-CNN, respectively. Additionally, the mAP was 5.5 percentage points higher. This indicates that the improved model 7 was better than the original model for the detection and recognition of tomato key organs.

**Table 2.** Model average precision and mean average precision.

| Model | Average Precision (%) | | | mAP (%) |
|---|---|---|---|---|
| | **Flower** | **Green Tomato** | **Red Tomato** | |
| Original Faster R-CNN | 76.8 | 88.4 | 90.4 | 85.2 |
| Improved model 7 | 90.5 | 90.8 | 90.9 | 90.7 |

It seems all of the improvements done on this paper mostly affected the detection of flowers and green tomatoes. The main reason for this phenomenon is that the stems and leaves of tomatoes in the picture are more seriously occluded from flowers and immature green tomatoes. Additionally, the color of stems and leaves is very similar to that of green tomatoes, while the color of red tomatoes is clearly contrasted with its background color. Therefore, the improvement in the paper is a higher recognition rate for flowers and green tomatoes.

*5.3. Memory Requirements and Detection Times of Models*

It can be seen from Table 3 that compared with the original Faster R-CNN model, the memory requirement and detection time of the improved model 7 were reduced by about 79% and 23%, respectively, and the detection accuracy had a large improvement. This is because residual blocks and global mean pooling were used in resnet-50 to reduce the amount of parameters and to increase the depth of the network. This indicates that the improved model reduced the model parameters and improved the precision of model detection on the basis of guaranteeing no additional test time, which lays a theoretical foundation for the subsequent implantation of the model into the embedded system, and the development of portable devices for accurate and real-time detection of tomato key organs.

**Table 3.** Model parameter quantity and detection time.

| Model | Model Memory Requirement (MB) | Average Testing Time (s) | FPS (Frames per Second) |
| --- | --- | --- | --- |
| Original Faster R-CNN | 546.9 | 0.095 | 10.5 |
| Improved model 7 | 115.9 | 0.073 | 13.7 |

### 5.4. Detection Performance of Model

In order to verify the actual field prediction performance of the optimal model in this paper, the key organs of tomato were tested at 9:00 a.m. on a sunny morning in the greenhouse. It can be seen from Table 1 and Figure 9 that the mAP of the improved model 7 for the detection of key organs of tomato was higher, reaching 90.7%. Additionally, the confidence of each object was basically over 0.99. The above performance showed that the model had good detection performance on tomato flowers, immature green tomatoes, and mature red tomatoes in the actual field background.

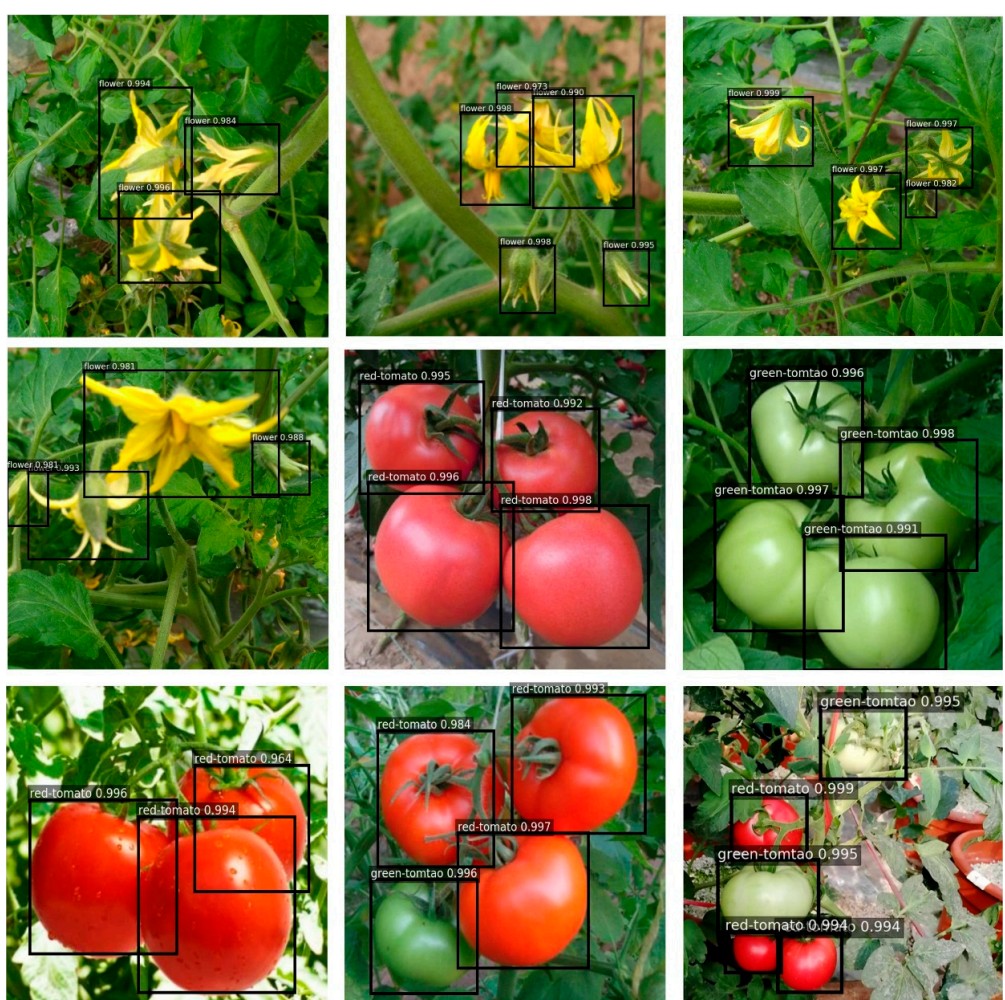

**Figure 9.** Examples of improved model detection. It can be seen that model 7, using the improved method of this paper has higher detection accuracy and confidence in the key organs of tomato under overlapping and occlusion, and that the detection performance on small targets is also good.

## 6. Conclusions

In this paper, CNN was used to detect and identify the key organs of tomato. The original Faster R-CNN model based on the vgg16 feature extraction network was improved, and the resnet-50 with residual structure was used to replace the vgg16 network. Additionally, K-means clustering was

adopted to fit the anchor size of the dataset in this paper, avoiding the problem of recognition accuracy reduction caused by artificial setting. Compared with the original Faster R-CNN model, this model used Soft-NMS to retain the generated detection boxes, which solved the problem of low detection accuracy of key tomato organs under overlapping and occlusion, to a certain extent.

The results shows that the APs of tomato flowers, immature green tomatoes, and mature red tomatoes were increased from 76.8%, 88.4%, 90.4% to 90.5%, 90.8%, 90.9%, respectively. Additionally, the memory required by the model was reduced from 546.9 MB to 115.9 MB. The average detection time was shortened from 0.095 S/sheet to 0.073 S/sheet. The mAP was increased from 85.2% to 90.7%, and the performance of the model was greatly improved.

The training model can be transplanted to the embedded system in the future, which lays a theoretical foundation for the development of a precise pesticide targeting application system, and an automatic tomato picking robot device.

**Author Contributions:** J.S. (Jun Sun) and X.H. contributed to the development of the systems, including farm site data collection, and the manuscript writing. X.H. provided significant suggestions on the development, and contributed to performance evaluation. X.G. and X.W. contributed to grammar modification. J.S. (Jifeng Shen), X.H., X.G. and Y.S. analyzed the results. All authors wrote the manuscript together.

**Funding:** This work is partially supported by National Natural Science Funds projects (31471413, 61875089), A Project Funded by the Priority Academic Program Development of Jiangsu Higher Education Institutions (PAPD), Six Talent Peaks Project in Jiangsu Province (ZBZZ-019), Science and Technology Support Project of Changzhou (Social Development) (CE20175042).

**Conflicts of Interest:** The authors declare no conflict of interest.

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
