# Peer review of "Detection of Key Organs in Tomato Based on Deep Migration Learning in a Complex Background"

_agriculture, doi:10.3390/agriculture8120196_

Round 1

Reviewer 1 Report

The manuscript proposes a novel methodology to detect key plant’s part in tomato cultivation. The methodology is based on deep learning algorithm with the main goal of improving the detection of those parts that result occluded.

The work presented seems quite interesting, however, many parts of the manuscript result quite difficult to follow. Many grammatical errors are also present. Therefore I will suggest the authors proofread the entire manuscript.

The abstract should be better structured, presenting the main problem the Authors want to solve and a summary of the proposed mythology, without including details that need further explanation.

The Introduction Section presents issues with reference. When the-state-of-the-art is analyzed, the pros and cons of the methodology should be stated, in order to understand if there is an advancement or not by the proposed technique.

Datasets used in this work, such as the VOC 2007 dataset, should be better described and referenced.

In general, the methodology result quite fragmented and difficult to follow. It is not clear where the novelty lies. Authors should clearly state that.

Equations are not explained and many variables are not defined.

Figures should be also better explained. Captions are quite minimal providing any useful information. It would be quite helpful to have a complete scheme of the proposed methodology, including the clustering part. 

I would also suggest to include the partial results along the process in order to show the various inputs and outputs. For example, the results after the k-mean.

Many parameters have been used but a discussion on their choice is missing. A critical analysis of the selected parameters should be provided.

It is not clear what detection effect means. I would suggest changing in “Detection under different condition”.

It is not clear how many runs have been performed to provide the mAP.

In general, the manuscript must be proofread and better structured. The text is quite unclear making the reading not easy. Methodology and parameter description must be improved and better described.

Author Response

Response to Reviewer 1 Comments

Point 1: The work presented seems quite interesting, however, many parts of the manuscript result quite difficult to follow. Many grammatical errors are also present. Therefore I will suggest the authors proofread the entire manuscript.

Response 1: Thank you for your comments. I have proofread the manuscript.

Point 2: The abstract should be better structured, presenting the main problem the Authors want to solve and a summary of the proposed mythology, without including details that need further explanation.

Response 2: Thank you for your comments. I have modified the summary section and removed some details.  

Point 3: The Introduction Section presents issues with reference. When the-state-of-the-art is analyzed, the pros and cons of the methodology should be stated, in order to understand if there is an advancement or not by the proposed technique.

Response 3: Thank you for your comments. I have modified them and at the end of the second paragraph of the introduction, some brief analysis of the above methods has been added.

Point 4: Datasets used in this work, such as the VOC 2007 dataset, should be better described and referenced.

Response 4: Thank you for your comments. I have added the sentence “labelimg software was used to rectangle the tomato flowers, green tomatoes and red tomatoes in the data set. After each image was annotated, it will generate a corresponding xml file containing the category and location information of the target, similar to the dataset format of pascal visual object classes challenge 2007” and added a citation.

Point 5: In general, the methodology result quite fragmented and difficult to follow. It is not clear where the novelty lies. Authors should clearly state that.

Response 5: Thank you for your comments. I have added some sentences.In this section, the method proposed by this paper was evaluated and analyzed on three experimental settings: (1) detection performance under different basic anchor size; (2) detection performance under different pooling type of basic feature extraction network; (3) detection performance under different retention algorithm of the bounding box.in chapter four.

Point 6: Equations are not explained and many variables are not defined.

Response 6: Thank you for your comments. I have updated.

Point 7: Figures should be also better explained. Captions are quite minimal providing any useful information. It would be quite helpful to have a complete scheme of the proposed methodology, including the clustering part.

Response 7: Thank you for your comments. I have explained the figures now.

Point 8: I would also suggest to include the partial results along the process in order to show the various inputs and outputs. For example, the results after the k-mean.

Response 8: Thank you for your comments. I have added some visualized results in the paper.

Point 9: Many parameters have been used but a discussion on their choice is missing. A critical analysis of the selected parameters should be provided.

Response 9: Thank you for your comments. I have added some analysis of the selected parameters.

Point 10: It is not clear what detection effect means. I would suggest changing in “Detection under different condition”.

Response 10: Thank you for your comments. I have changed the subtitle of chapter 4 and removed thedetection effect.

Point 11: It is not clear how many runs have been performed to provide the mAP.

Response 11: Thank you for your comments. I have added the total number of runs.

All changes are displayed in the latest submitted manuscript, please check the updated manuscript.

Reviewer 2 Report

The paper presents a new CNN based method to identify key organs in tomato. This research topic is interesting and fit the scope of this journal. The overall, the analysis of results and the conclusions are solid. In my opinion, the study contributes to the advancement of AI knowledge and its applications in agriculture. However, I propose several comments to improve the manuscript.

PAG 3 LINE 104. Please, give details about the camera and the size of the original images.

PAG 3 LINE 110. It is always difficult to obtain a sufficient number of real images for the training stage in deep learning, so authors have also generated some synthetic tomato images for training the network. Some comment about the trouble generalizing to real data because of the change in image statistics?

FORMAL ASPECTS

Abbreviations and acronyms are often defined the first time they are used within the abstract and again in the main text and then used throughout the remainder of the manuscript. Please consider adhering to this convention (e.g. CNN, defined in PAGE 2 LINE 62, it is used arbitrarily henceforth)

PAG 1 LINE 20. A space is missing before “MB”,

PAG 5 LINE 39. A space is missing after “to”

      PAG 13 LINE 392. Correct: “stress” markers

PAG 13 LINE 400. Correct: “Computer Vision”

LANGUAGE

The language of the manuscript is clear, though it has some grammatical errors and some sentences need to be clarified and others can be shortened.

Author Response

Response to Reviewer 2 Comments

Point 1: PAG 3 LINE 104. Please, give details about the camera and the size of the original images.

Response 1: Thank you for your comments. I have added some details about the camera and the size of the images.

Point 2: PAG 3 LINE 110. It is always difficult to obtain a sufficient number of real images for the training stage in deep learning, so authors have also generated some synthetic tomato images for training the network. Some comment about the trouble generalizing to real data because of the change in image statistics?

Response 2: Thank you for your comments. A large amount of internet information showed that a lot of data was required to train models by using deep learning methods. Data augmentation techniques are often used in deep learning in order to enhance the robustness of the model and reduce the over-fitting that occurs during training models.

Point 3: Abbreviations and acronyms are often defined the first time they are used within the abstract and again in the main text and then used throughout the remainder of the manuscript. Please consider adhering to this convention (e.g. CNN, defined in PAGE 2 LINE 62, it is used arbitrarily henceforth)

Response 3: Thank you for your comments. I have modified them.

Point 4: PAG 1 LINE 20. A space is missing before “MB”,

Response 4: Thank you for your comments. I have added a space before “MB”. But another expert suggested that I need to streamline the abstract, I deleted this sentence.

Point 5: PAG 5 LINE 39. A space is missing after “to”.

Response 5: Thank you for your comments. I have added a space after “to”

Point 6: PAG 13 LINE 392. Correct: “stress” markers

Response 6: Thank you for your comments. I have updated the citation.

Point 7: PAG 13 LINE 400. Correct: “Computer Vision”

Response 7: Thank you for your comments. But I can't find “Computer Vision” in PAG 13 LINE 400.

Point 8: The language of the manuscript is clear, though it has some grammatical errors and some sentences need to be clarified and others can be shortened.

Response 8: Thank you for your comments. I have updated the paper. This is my first time writing an English paper and I will pay attention to it later.

All changes are displayed in the latest submitted manuscript, please check the updated manuscript.

Reviewer 3 Report

General Comment:

The authors improved a model to detect tomato within RGB images in terms of the accuracy and the speed rate. In the introduction, a strong story has been addressed to show the importance of developing and improving detection/classification/segmentation methods. The pros and cons of the existing methods are highlighted. The only thing that I thought needs to be added here is to address the effect of shaded pixels on misclassification that it may happen in this type of studies. In addition, there is a technical issue to link the literature reviews to the reference section that must be solved in the next round. Also, It is not necessary to mention the first name of authors in this section. The methodology section is fine and is correctly designed. I have listed some minor issues regarding the definition of variables, improving the quality of figures and highlighting a major issue for training the model. For the results section, I have some concerns for the indices reported to show the performance of the improved version of the detection model versus the traditional one. Also, some informative description must be added there. I have listed my comments/ suggestions in the following lines.

   Introduction:

(1)    Regarding the name of authors in the introduction, mentioning the last name is enough. The authors have to remove the first name of the authors mentioned for the literature reviews.

(2)    Line 44 “Traditional tomato recognition methods mostly extract the information of color, shape and some shallow features” the shallow features must be defined. What does it mean? If it is possible, mention an example for that.

(3)    Line 64- 65: “It can not only learn shallow semantic information, but also learn deep abstract information” it requires a proper citation.

(4)    Line 68: “Faster R-CNN and YOLO models have been widely used and achieved good result”. As a rule of thumb, any abbreviation must be defined at the first appearance. Check YOLO, R-CNN, Soft-NMS etc.

(5)    Line 86: “So, It is feasible to use deep learning methods to detect…” Change “So” to “Thus” or “Therefore”.

(6)    Line 87-88: “However, the accuracy of these models is not high under the conditions of overlapping and occlusion ”. First of all, it requires a proper citation. Second, the overlapping and occlusion must be clearly described here. After reading the introduction, these conditions don’t make sense to me.

    Materials and Methods:

(1)    Figure 1: It seems the scale of some of the sub figures is not correct (some of them are wider and some of them are taller see the top right and the middle left).

(2)    Line 103-104: “A total of 5624 RGB images of tomato flowers…. were collected…. With high-definition camera”. I highly recommend adding a table to show the details information of the camera, the date and time of capturing images, the resolution size, etc.

(3)    Line 113: “The total number of expanded samples was 8929…” Something here is confusing. In Line 102 the authors mentioned that “A total of 5624 RGB images of …were collected” I’m wondering to know how it could be possible to have 8929 samples from 5624 images.

(4)    Line 116: “2.2 Structure of the detection model of tomato organs” I think the information mentioned in this sub-section is just related to the structure of detection models using CNN. So, it is better to remove “of tomato organs” from the title.

(5)    Line 128: “The traditional Faster R-CNN object detection algorithm can be divided into two parts.” Including what?

(6)    Figure 2: What is “Relu” in this figure

(7) Figure 3: What is “FC” in this figure? I highly recommend using the simple arrows in this figure.

(8) Eq. (2): IOU stands for what?

(9) I’m wondering to know how the authors trained the model particularly for detecting the tomato. Do they label the red tomato manually or used the existing datasets?  

Results and Analysis:

(1)    Table (1): In general, the results reported here show that the assumption of the authors in terms of improving the accuracy of the model with changing the anchor size through KNN works very well (mAP increased from 85% to 90%). However, my question here is related to the index that is reported to show the performance of the model. I’m wondering to know if mAP is calculated based on the testing datasets or both (training and testing). Because if it is reported for both, it is hard to make a reliable decision to infer the performance of the model is improved. If it is reported for both, I highly recommend adding the mAP for testing datasets as well. In this situation, we can see whether the K-means clustering could improve the model in terms of testing datasets or it led to increasing the accuracy just in training datasets (probably overfitting happened). Since this table is a core of the results and it is hard to follow the impact of anchor size, pooling type, and detection box, if it is possible, it is better to be converted into a figure like a hierarchy graph.

(2)    Line 267-268: “The results show that the effect of maximum pooling is better than the average pooling …” This is good to evaluate a model in terms of the type of modules involved in the model. Nevertheless, I want to raise this question: This conclusion could be a function of the datasets or not? Could be sensitive to the object that is detected (i.e. tomato here)? Can we suggest to other researchers to use the maximum pooling instead of the average pooling?

(3)    Figure 4 and Figure 6: I do not know how difficult it could be but if it is possible, change the red box around the tomato to the black box and increase the line width. Also, change the blue background to the black but keep the white font color and make sure that the information mentioned here is readable (Probably it is better to use a figure with 600 or 1200 dpi).

(4)    Figure 5: limit the y-axis from 0.6 for both sub-images. Also, add an informative description of this figure.

(5)     Table 2: It seems all the improvements done on R-CNN mostly affect the detection of flower and green tomato. I’m wondering to know if there is a scientific description (or mathematic reason) to justify this enhancement.  

Author Response

Response to Reviewer 3 Comments

Point 1: (1) Regarding the name of authors in the introduction, mentioning the last name is enough. The authors have to remove the first name of the authors mentioned for the literature reviews.

Response 1: Thank you for your comments. I have removed the first name of the authors mentioned for the literature reviews.

Point 2: (2) Line 44 “Traditional tomato recognition methods mostly extract the information of color, shape and some shallow features” the shallow features must be defined. What does it mean? If it is possible, mention an example for that.

Response 2: Thank you for your comments. I have added “that is the visual features of the bottom layer, such as texture features and the colors and shapes mentioned earlier”.

Point 3: (3) Line 64- 65: “It can not only learn shallow semantic information, but also learn deep abstract information” it requires a proper citation.

Response 3: Thank you for your comments. Sorry, I forgot to insert a reference symbol here, and I have added it.

Point 4: (4) Line 68: “Faster R-CNN and YOLO models have been widely used and achieved good result”. As a rule of thumb, any abbreviation must be defined at the first appearance. Check YOLO, R-CNN, Soft-NMS etc.

Response 4: Thank you for your comments. I have checked them.

Point 5: (5) Line 86: “So, It is feasible to use deep learning methods to detect…” Change “So” to “Thus” or “Therefore”.

Response 5: Thank you for your comments. I have changed it.

Point 6: (6) Line 87-88: “However, the accuracy of these models is not high under the conditions of overlapping and occlusion”. First of all, it requires a proper citation. Second, the overlapping and occlusion must be clearly described here. After reading the introduction, these conditions don’t make sense to me.

Response 6: Thank you for your comments. First of all, I think this sentence is a summary of the shortcomings of the methods used in the above literature, so you may not have to add a citation. And I also added some explanations in the above literature. Secondly, I added a brief description of the overlapping and occlusion.

Point 7: (7) Figure 1: It seems the scale of some of the sub figures is not correct (some of them are wider and some of them are taller see the top right and the middle left).

Response 7: Thank you for your comments. I have updated Figure 1.

Point 8: (8) Line 103-104: “A total of 5624 RGB images of tomato flowers…. were collected…. With high-definition camera”. I highly recommend adding a table to show the details information of the camera, the date and time of capturing images, the resolution size, etc.

Response 8: Thank you for your comments. I have added “These pictures were taken at 9:00 am, 12:00am, and 3:00 pm, July-August, 2018. And the camera is a CMOS sensor, which is the camera that comes with the honor 9 mobile phone.”

Point 9: (9) Line 113: “The total number of expanded samples was 8929…” Something here is confusing. In Line 102 the authors mentioned that “A total of 5624 RGB images of …were collected” I’m wondering to know how it could be possible to have 8929 samples from 5624 images.

Response 9: Thank you for your comments. “In this paper, python script was used to augment a small number of sample maps, including random flip (horizontal, vertical), transform angle (0 ~ 180), random scaling of the original image scaling factor (1 ~ 1.5) etc.” This sentence in the article explains that I have used the data augmentation method of random flip, rotation angle and random scaling on 5,624 original images to expand the image separately, and expanded the image to 8929.

Point 10: (10) Line 116: “2.2 Structure of the detection model of tomato organs” I think the information mentioned in this sub-section is just related to the structure of detection models using CNN. So, it is better to remove “of tomato organs” from the title.

Response 10: Thank you for your comments. I have removed “of tomato organs” from the title.

Point 11: (11) Line 128: “The traditional Faster R-CNN object detection algorithm can be divided into two parts.” Including what?

Response 11: Thank you for your comments. I have added “One is the feature extraction layer and the other is the region proposal network.”

Point 12: (12) Figure 2: What is “Relu” in this figure

Response 12: Thank you for your comments. ReLU (Rectified Linear Units) is the activation function in CNN, and I have added some information about “ReLU” in Figure 2.

Point 13: (13) Figure 3: What is “FC” in this figure? I highly recommend using the simple arrows in this figure.

Response 13: Thank you for your comments. “FC” is the fully connected layer in CNN, I have added some information about “FC” in Figure 3 and used the simple arrows.

Point 14: (14) Eq. (2): IOU stands for what?

Response 14: Thank you for your comments. I have updated the chapter 2.4 for more details and added a brief definition of IOU. You can also browse the literature 22 for more information.

Point 15: (15) I’m wondering to know how the authors trained the model particularly for detecting the tomato. Do they label the red tomato manually or used the existing datasets? 

Response 15: Thank you for your comments. Training the model for detecting the tomato is just as shown in Figure 3. Firstly, a feature extraction network was used to extract the features of key organs of tomato. Then, object classification was carried out on the generated feature map, and the bounding boxes were generated by using RPN (region proposal network). Finally, the two layers are output to the fully connected layer to complete the identification and localization tasks.

Point 16: (16) Table (1): In general, the results reported here show that the assumption of the authors in terms of improving the accuracy of the model with changing the anchor size through KNN works very well (mAP increased from 85% to 90%). However, my question here is related to the index that is reported to show the performance of the model. I’m wondering to know if mAP is calculated based on the testing datasets or both (training and testing). Because if it is reported for both, it is hard to make a reliable decision to infer the performance of the model is improved. If it is reported for both, I highly recommend adding the mAP for testing datasets as well. In this situation, we can see whether the K-means clustering could improve the model in terms of testing datasets or it led to increasing the accuracy just in training datasets (probably overfitting happened). Since this table is a core of the results and it is hard to follow the impact of anchor size, pooling type, and detection box, if it is possible, it is better to be converted into a figure like a hierarchy graph.

Response 16: Thank you for your comments. I am sorry that I forgot to explain this. The mAP is calculated based on the testing datasets. And I have changed it in Table (1). Table 1 was visualized to more intuitively show the mean average precision of the models with different parameter settings.

Point 17: (17) Line 267-268: “The results show that the effect of maximum pooling is better than the average pooling …” This is good to evaluate a model in terms of the type of modules involved in the model. Nevertheless, I want to raise this question: This conclusion could be a function of the datasets or not? Could be sensitive to the object that is detected (i.e. tomato here)? Can we suggest to other researchers to use the maximum pooling instead of the average pooling?

Response 17: Thank you for your comments. This conclusion may not be a function of the datasets. Average pooling retains more background information of the image, and maximum pooling retains more texture information. In the object detection task, the texture information may need to be more, and the interference information such as the background may be reduced as much as possible. And most target detection algorithms also use maximum pooling. So we can suggest to other researchers to use the maximum pooling instead of the average pooling in object detection.

Point 18: (18) Figure 4 and Figure 6: I do not know how difficult it could be but if it is possible, change the red box around the tomato to the black box and increase the line width. Also, change the blue background to the black but keep the white font color and make sure that the information mentioned here is readable (Probably it is better to use a figure with 600 or 1200 dpi).

Response 18: Thank you for your comments. I have updated the figure as your request.

Point 19: (19) Figure 5: limit the y-axis from 0.6 for both sub-images. Also, add an informative description of this figure.

Response 19: Thank you for your comments. Thank you for your comments. I have updated the figure as your request and added an informative description of this figure.

Point 20: (20) Table 2: It seems all the improvements done on R-CNN mostly affect the detection of flower and green tomato. I’m wondering to know if there is a scientific description (or mathematic reason) to justify this enhancement.

Response 20: Thank you for your comments. The main reason for this phenomenon is that the stems and leaves of tomatoes in the picture are more seriously occluded from flowers and immature green tomatoes, and the color of stems and leaves is very similar to that of green tomatoes, while the color of red tomatoes is clearly contrasted with its background color, so the improvement in the article has a higher recognition rate for flowers and green tomatoes. I also added the explanation in page 5.2.

All changes are displayed in the latest submitted manuscript, please check the updated manuscript.

Reviewer 4 Report

The paper presents a modification to faster-rcnn in order to detect key organs in tomato plants. To my understanding the main contributions of the paper are the use of ResNet instead of VGG for feature detection, k-means to adapt anchor size selection and soft-nms. All three contributions seem minor and not sufficiently novel for publication. Furthermore my main concerns to the methodology of the paper are:

1. The use of k-means to select anchor size is not clearly explained and I am still doubtful of how it was used to obtain pixel size.

2. Authors claim to use data augmentation but then mention (line 113) that the dataset increases from 5624 to 8929. How is that? Isn't data augmentation applied randomly while training (e.g. random rotations)?

3. Authors test with ResNet50 but no explanation is done to exclude other architectures such as ResNet101 or ResNet152.

4. Training explains the initialisation of  convolutional weights. Does that mean that ResNet is trained from scratch? 

5. Results need to be re-organised as mAP is given and then explained much later. 

Author Response

Response to Reviewer 4 Comments

Point 1: (1) The use of k-means to select anchor size is not clearly explained and I am still doubtful of how it was used to obtain pixel size.

Response 1: Thank you for your comments. I have updated the chapter 2.4 for more details, and you can browse the literature 23 for more information.  

Point 2: (2) Authors claim to use data augmentation but then mention (line 113) that the dataset increases from 5624 to 8929. How is that? Isn't data augmentation applied randomly while training (e.g. random rotations)?

Response 2: Thank you for your comments. I think you may not understand what I mean. I collected a total of 5,624 RG images, and then used python scripts to augment these images, such as random flip, rotation angle and random zoom size. The augmented picture became a total of 8,929 sheets for model training and testing.

Point 3: (3) Authors test with ResNet50 but no explanation is done to exclude other architectures such as ResNet101 or ResNet152.

Response 3: Thank you for your comments. Sorry, I don't think about it carefully. ResNet101 and ResNet152 may be a little better than ResNet50, but it takes a long time to train the pre-training model from scratch. And I think the effect of changing the network structure from VGG16 to ResNet50 has been greatly improved, and it is not necessary to use the ResNet101 or ResNet152 model. And the memory of GPU is not enough to train the ResNet-152 with batch normalization.

Point 4: (4) Training explains the initialisation of convolutional weights. Does that mean that ResNet is trained from scratch?

Response 4: Thank you for your comments. Yes, Resnet50 was trained from scratch. Firstly, we used the Image Net dataset to initialize it, and then migrated the pre-trained model to the Faster R-CNN framework.

Point 5: (5) Results need to be re-organised as mAP is given and then explained much later.

Response 5: Thank you for your comments. I have modified the structure of the paper.

All changes are displayed in the latest submitted manuscript, please check the updated manuscript.

Round 2

Reviewer 3 Report

All of my comments are answered by the authors. However, there are several minor issues particularly grammatical issues in the modified version of the manuscript that need to be considered by the authors. The authors are strongly recommended to ask an English partner to edit this version of the manuscript.

Line 18-19: “The test results showed that compared with the traditional Faster R-CNN model, the mean average precision (mAP) was significantly improved”. Change it to “The test results showed that the mean average precision (mAP) was significantly improved compared with the traditional Faster R-CNN model,”

Line 46-47: “used the combination of Harr features and Adaboost classifier to identify tomatoes” What are Harr features? The Harr features must be defined in the manuscript.

Line 50-51: “However, the recognition rate of immature green tomato which the color is similar to the color of the leaf is not high” Check this sentence. “which” or “in which?”

Line 57-58: “But this method has a poor identification”. Change “has” to “had”

Line 75-76: “and difficult to understand”. Remove it.

Line 77-78: “but the model parameter was too large and was not suitable for real-time detection” Change it to “but the number of model parameters was too large and was not suitable for real-time detection”

Line 89-91: “(The will be overlapping and occlusion between the tomatoes, and the stems and leaves of the tomato will also cover in the fruit).” Remove parenthesis.

Line 118-119: “In this paper, labelimg 118 software was used to label the tomato flowers”. labelimg or labeling software? Also mention a reference for that software (e.g. url, …)

Line 120-121: “it will generate a corresponding xml file containing” change it to “it generates a corresponding xml file containing”

Line 124-125: “python script was used to augment a small number” Change it to “a python script was used to augment a small number”

Line 159: “the problem of precision decline after a certain iteration still exists”. Change it to “the problem of precision decline after a certain iteration remains”

The caption of Figure 2. “Residual learning module. Since direct mapping is difficult to learn, the basic mapping relationship from X to H(x) is no longer being learned, but the difference between the two is learned, that is the residual, then in order to calculate H(x), just add this residual to the input.” What is H(x)? H(x) or F(x)? H(x) must be shown in the figure.

Line 297-268: “the method proposed by this paper was evaluated and analyzed on three experimental settings” Change it to “the performance of the proposed method was evaluated and analyzed in three experimental settings”

Line 318: “So K-means need to be used to cluster the appropriate anchors” Change it to “Thus, K-means need to be used to cluster the appropriate anchors.”

Figure 4: Change the x-axis label to “Model NO.”. Remove Legend. Bold the numbers. This figure is not a histogram. It is a bar graph or a bar chart. Therefore, choose a proper title (i.e. The bar chart of mAP for different model parameter settings).

Figure 6: “It can be seen from the figure that the model using the maximum pooling has higher confidence than the model using the average pooled model, and some targets similar to the colors of the background are well recognized. This is because the maximum pooling retains more texture information, while the average pooling retains more background information.” Remove this part from the caption and mention it in the manuscript.

Line 357-358: “It can be seen from Table 1”. Change it to “It can be seen from Table 1 and Figure 4”

Line 357-359: “It can be seen from Table 1 that the mAP of model 5, model 6, model 8 and model 7 retained by the using Soft-NMS method are 2.3, 2.3, 2 and 2 percentage points higher than those of the models 1, 2, 3 and 4 retained by the using NMS method, respectively.” Re-write it. It is not clear.

Figure 7: “It can be seen that because the dataset in this paper has the overlapping between tomatoes and the occlusion of the stems and leaves, the model 7 using Soft-NMS achieve a better result. The main reason for this is that model 7 is based on the area of overlapping portion to set an attenuation function for the adjacent detection box instead of completely zeroing its score.” Remove this part from the caption and mention it in the manuscript.

Figure 8: “These two figures show the relationship between recall and precision. The area under the curve represents the AP of the class, and the larger the area, the higher the AP. For a detailed introduction to the P-R curve, we can refer to [29]” Remove this part from the caption and mention it in the manuscript.

Line 435-436: “So the improvement in the paper has a higher recognition rate for flowers 435 and green tomatoes.” Change it to “Therefore, the improvement has a higher recognition rate for flowers and green tomatoes”

Table 3: What is FPS. Must be defined.

Author Response

Response to Reviewer 3 Comments

Point 1: (1) Line 18-19: “The test results showed that compared with the traditional Faster R-CNN model, the mean average precision (mAP) was significantly improved”. Change it to “The test results showed that the mean average precision (mAP) was significantly improved compared with the traditional Faster R-CNN model,”

Response 1: Thank you for your comments. I have changed it to “The test results showed that the mean average precision (mAP) was significantly improved compared with the traditional Faster R-CNN model,”

Point 2: (2) Line 46-47: “used the combination of Harr features and Adaboost classifier to identify tomatoes” What are Harr features? The Harr features must be defined in the manuscript.

Response 2: Thank you for your comments. I have added some details about the Harr features.

Point 3: (3) Line 50-51: “However, the recognition rate of immature green tomato which the color is similar to the color of the leaf is not high” Check this sentence. “which” or “in which?”

Response 3: Thank you for your comments. Sorry, it’s my fault. I have changed “which” into “in which”.

Point 4: Line 57-58: “But this method has a poor identification”. Change “has” to “had”

Response 4: Thank you for your comments. I have changed “has” to “had”.

Point 5: Line 75-76: “and difficult to understand”. Remove it.

Response 5: Thank you for your comments. I have removed it.

Point 6: Line 77-78: “but the model parameter was too large and was not suitable for real-time detection” Change it to “but the number of model parameters was too large and was not suitable for real-time detection”

Response 6: Thank you for your comments. I have changed it to “but the number of model parameters was too large and was not suitable for real-time detection”.

Point 7: Line 89-91: “(The will be overlapping and occlusion between the tomatoes, and the stems and leaves of the tomato will also cover in the fruit).” Remove parenthesis.

Response 7: Thank you for your comments. I have removed the parenthesis.

Point 8: Line 118-119: “In this paper, labelimg 118 software was used to label the tomato flowers”. labelimg or labeling software? Also mention a reference for that software (e.g. url, …)

Response 8: Thank you for your comments. I wrote the word wrong. I have corrected it and added links.

Point 9: Line 120-121: “it will generate a corresponding xml file containing” change it to “it generates a corresponding xml file containing”

Response 9: Thank you for your comments. I have changed it to “it generates a corresponding xml file containing”.

Point 10: Line 124-125: “python script was used to augment a small number” Change it to “a python script was used to augment a small number”

Response 10: Thank you for your comments. I have changed it to “a python script was used to augment a small number”.

Point 11: Line 159: “the problem of precision decline after a certain iteration still exists”. Change it to “the problem of precision decline after a certain iteration remains”

Response 11: Thank you for your comments. I have changed it to “the problem of precision decline after a certain iteration remains”.

Point 12: The caption of Figure 2. “Residual learning module. Since direct mapping is difficult to learn, the basic mapping relationship from X to H(x) is no longer being learned, but the difference between the two is learned, that is the residual, then in order to calculate H(x), just add this residual to the input.” What is H(x)? H(x) or F(x)? H(x) must be shown in the figure.

Response 12: Thank you for your comments. I have updated it on figure 2.

Point 13: Line 297-268: “the method proposed by this paper was evaluated and analyzed on three experimental settings” Change it to “the performance of the proposed method was evaluated and analyzed in three experimental settings”

Response 13: Thank you for your comments. I have changed it to “the performance of the proposed method was evaluated and analyzed in three experimental settings”.

Point 14: Line 318: “So K-means need to be used to cluster the appropriate anchors” Change it to “Thus, K-means need to be used to cluster the appropriate anchors.”

Response 14: Thank you for your comments. I have changed it to “Thus, K-means need to be used to cluster the appropriate anchors.”

Point 15: Figure 4: Change the x-axis label to “Model NO.”. Remove Legend. Bold the numbers. This figure is not a histogram. It is a bar graph or a bar chart. Therefore, choose a proper title (i.e. The bar chart of mAP for different model parameter settings).  

Response 15: Thank you for your comments. I have modified Figure 4 as you requested.

Point 16: Figure 6: “It can be seen from the figure that the model using the maximum pooling has higher confidence than the model using the average pooled model, and some targets similar to the colors of the background are well recognized. This is because the maximum pooling retains more texture information, while the average pooling retains more background information.” Remove this part from the caption and mention it in the manuscript.

Response 16: Thank you for your comments. I have removed this part from the caption and mention it in the manuscript.

Point 17: Line 357-358: “It can be seen from Table 1”. Change it to “It can be seen from Table 1 and Figure 4”

Response 17: Thank you for your comments. I have changed it to “It can be seen from Table 1 and Figure 4”.

Point 18: Line 357-359: “It can be seen from Table 1 that the mAP of model 5, model 6, model 8 and model 7 retained by the using Soft-NMS method are 2.3, 2.3, 2 and 2 percentage points higher than those of the models 1, 2, 3 and 4 retained by the using NMS method, respectively.” Re-write it. It is not clear.

Response 18: Thank you for your comments. I have rewritten it.

Point 19: Figure 7: “It can be seen that because the dataset in this paper has the overlapping between tomatoes and the occlusion of the stems and leaves, the model 7 using Soft-NMS achieve a better result. The main reason for this is that model 7 is based on the area of overlapping portion to set an attenuation function for the adjacent detection box instead of completely zeroing its score.” Remove this part from the caption and mention it in the manuscript.

Response 19: Thank you for your comments. I have removed this part from the caption and mentioned it in the manuscript.

Point 20: Figure 8: “These two figures show the relationship between recall and precision. The area under the curve represents the AP of the class, and the larger the area, the higher the AP. For a detailed introduction to the P-R curve, we can refer to [29]” Remove this part from the caption and mention it in the manuscript.

Response 20: Thank you for your comments. I have removed this part from the caption and mentioned it in the manuscript.

Point 21: Line 435-436: “So the improvement in the paper has a higher recognition rate for flowers 435 and green tomatoes.” Change it to “Therefore, the improvement has a higher recognition rate for flowers and green tomatoes”

Response 21: Thank you for your comments. I have changed it to “Therefore, the improvement has a higher recognition rate for flowers and green tomatoes”.

Point 22: Table 3: What is FPS. Must be defined.

Response 22: Thank you for your comments. I have added an explanation of FPS in table 3. Thank you for your patience. All changes are displayed in the latest submitted manuscript, please check the updated manuscript.

Reviewer 4 Report

Point 1: I am still confused by the explanation of the use of k-means to select anchor box size. If I understand correctly, the authors claim to re-use the proposal in 23 to select anchor size in their Faster-RCNN implementation. However, it is still not clear what k is used and what values of k were explored. Finally, what is the effect of this clustering in the bounding box sizes 8 * 8, 16 * 16, 32 * 32? Please further explain this.

Point 2: In my opinion, using random flips, rotations and zoom size, and only augmenting the data offline less than twofold just misses the benefits of inline data augmentation. Authors should prove (by at least reporting an experiment) of the gains using no data augmentation, offline data augmentation and online data augmentation.

Point 3,4,5: OK.

Author Response

Response to Reviewer 4 Comments

Point 1: I am still confused by the explanation of the use of k-means to select anchor box size. If I understand correctly, the authors claim to re-use the proposal in 23 to select anchor size in their Faster-RCNN implementation. However, it is still not clear what k is used and what values of k were explored. Finally, what is the effect of this clustering in the bounding box sizes 8 * 8, 16 * 16, 32 * 32? Please further explain this.

Response 1: Thank you for your comments. First of all, I apologize for my fault. I made the anchor size wrong and I have updated it. I have added this sentence“The original Faster R-CNN used 9 kinds of anchor size, so the value of k must be set to 9 to cluster to get the same number of new anchor size” to explain the value of k. Secondly, anchor size is equal to the basic size multiplied by 16 pixels. As k-means algorithm was used to traverse the label information of all marked tomato key organs, the more appropriate anchor sizes suitable for the dataset were obtained according to the size of the manually labeled bounding box. When the clustering algorithm was run, new anchor sizes were generated (sizes from 69 pixels to 247 pixels), and the original anchor size 8*16, 16*16, 32*16 pixels was updated to 4*16, 8*16, 16*16 pixels. So that RPN could better detect some small-size targets when generating region proposals. It can avoid missing detection and improve recognition accuracy.

Point 2: In my opinion, using random flips, rotations and zoom size, and only augmenting the data offline less than twofold just misses the benefits of inline data augmentation. Authors should prove (by at least reporting an experiment) of the gains using no data augmentation, offline data augmentation and online data augmentation.

Response 2: Thank you for your comments. The algorithm used in this paper does not automatically expand the data during training. After your suggestion, I added the mAP of the dataset that did not use the augmentation in Table 1. Due to time constraints, all the parameter settings and analysis in this paper are performed on the dataset using offline data augmentation. Thank you for your patience. All changes are displayed in the latest submitted manuscript, please check the updated manuscript.
